# Connecting the Dots: AMOG/β_2_ and Its Elusive Adhesion Partner in CNS

**DOI:** 10.3390/ijms26178744

**Published:** 2025-09-08

**Authors:** Liora Shoshani, Christian Sosa Huerta, María Luisa Roldán, Arturo Ponce, Marlet Martínez-Archundia

**Affiliations:** 1Department of Physiology, Biophysics, and Neurosciences, Center for Research and Advanced Studies (Cinvestav), Mexico City 07360, Mexico; christian.sosa@cinvestav.mx (C.S.H.); mroldan@fisio.cinvestav.mx (M.L.R.); arturo.ponce@cinvestav.mx (A.P.); 2Laboratorio de Diseño y Desarrollo de Nuevos Fármacos e Innovación Biotecnológica (Laboratory for the Design and Development of New Drugs and Biotechnological Innovation), Sección de Estudios de Posgrado e Investigación, Escuela Superior de Medicina, Instituto Politécnico Nacional, Mexico City 11340, Mexico; mtmartineza@ipn.mx

**Keywords:** AMOG/β_2_, Na^+^/K^+^-ATPase, CAMs, neuron-glia interaction, glioblastoma

## Abstract

AMOG/β_2_, the β_2_ isoform of the sodium pump (Na^+^/K^+^-ATPase), functions as an adhesion molecule on glial cells, mediating critical neuron–astrocyte interactions during central nervous system (CNS) development. Despite its established role in glial adhesion, the neuronal receptor that partners with AMOG/β_2_ remains unknown. This review examines the structural and functional properties of AMOG/β_2_, including its capacity to form trans-dimers, both homophilic and potentially heterophilic—drawing comparisons with the β_1_ subunit, a well-characterized adhesion molecule. By integrating computational modeling, in vitro data, and structural predictions, we explore how factors such as N-glycosylation and cis-membrane interactions influence β_2_-mediated adhesion. We further consider candidate neuronal partners, including TSPAN31 and RTN4, and speculate on their potential roles in mediating heterophilic AMOG/β_2_ interactions. Finally, we discuss the broader implications of AMOG/β_2_ in neuron–glia communication, synaptic organization, neurodevelopment, and CNS disorders such as glioblastoma. Identifying the binding partner of AMOG/β_2_ holds promise not only for understanding the molecular basis of CNS adhesion but also for uncovering novel mechanisms of neuroglial regulation in health and disease.

## 1. Introduction

The central nervous system (CNS), comprising the brain and spinal cord, is crucial for processing information and regulating bodily functions. Interactions among neurons, glial cells (astrocytes, oligodendrocytes, and microglia), and endothelial cells of the blood–brain barrier are essential for CNS structure and function. Neurons communicate through synapses, transmitting electrical signals critical for brain activity, learning, and memory [1]. Glial cells, particularly astrocytes, regulate neurotransmitter levels, maintain the blood–brain barrier, and support neuronal metabolism [2].

During neurogenesis, glial cells provide scaffolding for neuronal migration and growth cones, offering guidance cues and potentially aiding neuronal proliferation [3]. Oligodendrocytes are responsible for myelination, enhancing nerve conduction speeds [4], while microglia act as immune cells, protecting the CNS from pathogens and clearing debris [5]. Cell–cell interactions are vital for synaptic plasticity, which underlies learning and memory [6,7]. These interactions are also critical for maintaining ionic balance, nutrient supply, and waste removal, all of which are essential for CNS homeostasis [8,9]. This complex network of interconnected cells relies on the expression of selective molecules known as cell adhesion molecules (CAMs), which mediate various forms of contact between neural cell surfaces. Notable families of CAMs include immunoglobulin superfamily cell adhesion molecules (IgCAMs), cadherins, integrins, and C-type lectin-like domain proteins (CTLDs).

Adhesion molecules are key in cell junctions throughout various tissues, including interneural synapses and neuron–glia junctions in the CNS. They help maintain tissue integrity through extracellular interactions and modulate intracellular signaling pathways important for cellular homeostasis. Altered expression of these molecules can disrupt signals that regulate cell growth, contributing to tumor formation. Understanding CAMs expands our knowledge of cellular interactions in neural tissues, revealing their roles in synapse formation and neuron–glia communication. The adhesion molecule on glia (AMOG) was first identified in 1987 [10] and was later found to be an isoform of the β-subunit of Na^+^/K^+^-ATPase (NKA), essential for maintaining sodium and potassium gradients across cell membranes [11,12,13]. The renaming to AMOG/β_2_ reflects its dual role in adhesion and ionic transport processes. The β-subunit serves as an accessory protein that aids in protein localization within the plasma membrane and K^+^ binding [14]. The two isoforms of the β-subunit, β_1_ and β_2_, act as adhesion molecules in epithelial tissues and the nervous system, respectively. In epithelial cells, β_1_ maintains intercellular junctions through homophilic β_1_-β_1_ adhesions, supporting cell polarity and transepithelial transport [15,16,17,18,19,20,21]. In the nervous system, β_2_ is primarily expressed in astrocytes, cerebellar granule cells, and photoreceptors, allowing NKA to function as both an ionic pump and a recognition molecule that mediates interactions between neurons and glial cells during brain development, promoting neurite outgrowth and cell migration [22].

Despite initial studies suggesting a heterophilic interaction between AMOG/β_2_ and a neural adhesion molecule (receptor), the specific pathways by which AMOG/β_2_ contributes to cell adhesion or the receptor molecule identity remain to be fully elucidated. This review examines the role of AMOG/β_2_ in the nervous system and poses the question: Does AMOG/β_2_ interact with an unidentified receptor on neurons?

## 2. Background

### 2.1. Adhesion Molecules in CNS

Adhesion molecules in the CNS are crucial for synapse formation, neuron–glia communication, and brain development. They facilitate cell–cell interactions, guiding neuronal migration and axon targeting. During synaptogenesis, adhesion molecules stabilize synaptic connections and regulate plasticity, which is essential for learning and memory. Disruptions in these molecules can lead to neurodevelopmental disorders, highlighting their importance in maintaining brain integrity and function [23,24]. Like epithelial junctions, cell–cell interactions in the CNS are mediated by various transmembrane proteins. These adhesion molecules can be found in interneural synapses and neuron–glia junctions, serving as important regulators of axon guidance and synapse formation [25]. We highlight key CAMs involved in synapse formation and astrocyte–synapse interactions, focusing on their homophilic and heterophilic characteristics.

#### 2.1.1. Classic Cadherins

Classic cadherins, such as N-cadherin, are calcium-dependent homophilic adhesion molecules critical for synaptic stability. They form symmetrical adhesion structures in synaptic junctions (puncta adherentia junctions) across most regions of the nervous system. Notably, classic cadherins exhibit binding specificity and region-specific distribution [26]. In the brain, various subtypes of classic cadherins are expressed by functionally connected nuclei and laminas [27,28].

#### 2.1.2. Protocadherins

This extensive family of cadherins is essential for establishing synaptic specificity. The clustered protocadherin (Pcdh) genes encode diverse cell-surface assemblies. Their combinatorial expression patterns generate the numerous address codes necessary for neuronal identity, allowing neurons to discriminate self from non-self, thereby contributing to the organization of neural circuits [29].

#### 2.1.3. Nectins

Nectins are a family of Ca^2+^-independent immunoglobulin (Ig)-like cell–cell adhesion molecules comprising four members (Nec1-Nec4). They form homo- or hetero-trans-dimers, with heterotypic binding resulting in stronger adhesion than homotypic binding. Nectins are integral to cell adhesion within synapses, particularly in puncta adherentia junctions (PAJs), and help cluster other adhesion molecules and signaling proteins to promote effective communication between neurons [30].

#### 2.1.4. Nectin-like Molecules (Necls)

Necl-2 acts as a homophilic adhesion molecule and exhibits heterophilic adhesion activity with Necl-1 and nectin-3. Like nectins, Necls facilitate interactions between neurons and supportive cells like astrocytes. Necl-2 localizes at synapses and is crucial for presynaptic differentiation and stabilization, which are essential for synapse formation and maintenance [31].

#### 2.1.5. NCAM (Neural Cell Adhesion Molecule)

The neural cell adhesion molecule (NCAM) contains five Ig-like domains and two fibronectin type III repeats, engaging in both homophilic and heterophilic interactions with various ligands at synapses, including fibroblast growth factor receptor (FGFR), L1, TAG-1/axonin-1, and heparan sulfate proteoglycans. NCAM is expressed widely in the developing and adult brain, playing critical roles in migration, axon pathfinding, and synaptic plasticity. It influences neuron–neuron and neuron–glia interactions, impacting early synaptogenesis and subsequent maturation [32,33].

#### 2.1.6. Integrins

Integrins are cell surface receptors that interact with the extracellular matrix (ECM) and transduce signals from the ECM to the cell. Comprising α- and β-subunits, integrins at the points of contact between neurons and astrocytes promote synaptogenesis. During CNS development, ECM receptors and their ligands serve as guidance molecules, informing neurons where and when to extend axons and dendrites, thus establishing synapses. Once stable synapses are formed, ECM receptors transition to regulate the maintenance of these connections and influence synaptic plasticity, with their activity being strongly affected by ECM composition [34].

#### 2.1.7. NgCAM (Neuron–Glia Cell Adhesion Molecule)

NgCAM facilitates adhesion between neurons and glial cells, contributing to nervous system development and the structure and signaling pathways of synapses.

#### 2.1.8. Contactins

Contactins (CNTNs) are a subfamily of the Ig superfamily of neural cell adhesion molecules (Ig-CAMs), consisting of six members (CNTN1-6) [35].

#### 2.1.9. TAG-1 (Transient Axonal Glycoprotein-1)

In the embryonic nervous system, Contactin-2/TAG-1 plays important roles in axonal elongation, axonal guidance, and cellular migration. In the postnatal nervous system, it also plays an essential role in the formation of myelinated nerve fibers [36].

#### 2.1.10. SYG-1 and SYG-2

These proteins, which are members of the Ig-SF, interact at the synapse, with Syg-1 being presynaptic and Syg-2 being postsynaptic. Their interactions induce the formation of synapses at appropriate targets, coordinating the assembly of synaptic components and ensuring connectivity [37,38].

#### 2.1.11. Sidekicks

Related to the immunoglobulin superfamily, sidekicks function as homophilic adhesion molecules in vitro and are highly concentrated at synapses in vivo. They play a role in specifying and stabilizing synaptic connections, which is crucial for neuronal communication [39].

#### 2.1.12. Neuroligins and Neurexins

Neuroligins are located on the postsynaptic side, interacting with presynaptic neurexins. β-Neurexin binds neuroligins trans-synaptically, inducing the formation of glutamatergic and GABAergic presynaptic specializations in vitro. This interaction influences synaptic strength and plasticity, which are essential for learning and memory [40,41].

### 2.2. The Na^+^/K^+^-ATPase in Neuron–Astrocyte Interactions

The molecule of interest, AMOG, corresponds to the β_2_ isoform of the Na^+^/K^+^-ATPase (NKA). NKA is a transmembrane heterodimer composed of a catalytic α subunit and an auxiliary β subunit, and may be further modulated by a third component from the FXYD family. The α subunit drives the enzyme’s fundamental activity, mediating the exchange of intracellular Na^+^ for extracellular K^+^, and thereby maintaining ionic gradients essential for cellular homeostasis. In contrast, the β subunit, which is heavily glycosylated, plays a critical role in regulating the assembly, maturation, and insertion of the α/β heterodimer into the plasma membrane, ensuring proper pump function and localization.

Beyond this structural role, β subunits also influence trafficking, expression, and pump activity: proper folding and maturation in the endoplasmic reticulum are critical for their assembly with α subunits [42]; β_2_ expression during kidney development correlates with apical localization of the pump in epithelial cells [43]; and N-glycosylation of β subunits modulates their polarized membrane distribution [44]. Recent studies also emphasize that β subunits can regulate NKA expression post-transcriptionally [45]. Among the β isoforms, β_2_ is unique in function: it stabilizes the Na^+^-occluded E1P state relative to the outward-open E2P state, an effect mediated by its transmembrane helix and not observed with β_1_ or β_3_, suggesting that its structural orientation directly tunes ion binding and pump activity [46].

Characterized in mammalian cells are four α, three β, and seven FXYD isoforms, demonstrating tissue-dependent distributions [47,48,49,50,51,52]. In the nervous system, the expression of NKA isoforms is complex, with neurons primarily producing the α_3_ polypeptide [53,54,55,56], and glial cells expressing α_2_ [57,58,59]. β isoforms also exhibit tissue-dependent distributions [59], with β_2_ found in skeletal muscle [60], the pineal gland [61], astrocytes [62], and granular cells [51], while β_3_ is present in the testis, retina, liver, and lung [63,64]. Expression patterns are influenced by developmental and hormonal regulation and can be altered in disease contexts [65,66,67,68,69].

AMOG/β_2_ functions as an adhesion molecule on glia and is expressed in mature brain astrocytes, focusing on interactions between astrocytes and other cell types. These bidirectional interactions play a crucial role in nervous system functioning. Astrocytes provide structural support to neurons and actively participate in synaptic modulation, neuronal development, and metabolism. They are involved in synapse formation and long-term plasticity related to learning [70,71,72,73], communicating with neural networks through gliotransmitters like gamma-aminobutyric acid (GABA) and adenosine triphosphate (ATP) to modulate neuronal activity and synaptic transmission [74,75]. Astrocytes also play a crucial role in regulating endothelial cells forming the blood–brain barrier (BBB), which protects the brain and controls the passage of substances between the brain and blood. Astrocytes regulate BBB permeability, nutrient supply, and waste removal [8,9,76]. NKA is essential in neuron–astrocyte interactions and astrocyte–vascular endothelium interactions. It maintains electrochemical gradients and participates in adhesion processes mediated by its β_2_ subunit [77]. Ongoing research aims to clarify how AMOG/β_2_ on astrocytes influences neuronal function and contributes to brain homeostasis, providing insights into the pathophysiology of neurological disorders and guiding therapeutic strategies to restore normal astrocytic function.

### 2.3. AMOG as a Heterophilic Adhesion Molecule

In 1987, Melitta Schachner’s group identified AMOG (Adhesion Molecule On Glia), a novel adhesion molecule distinct from known neural cell adhesion molecules [10]. This ~50 kD glycosylated integral membrane protein, primarily expressed by astrocytes, facilitates cerebellar granule cell migration by ensuring contact with Bergmann glial fibers. After migration, AMOG is prominently expressed in the internal granular layer, suggesting it may halt granule cell movement there [10,62,74].

Monoclonal AMOG antibodies showed specificity for glial surfaces during granule cell migration, indicating that AMOG does not participate in homophilic binding with neurons [10]. This was supported by findings that AMOG antibodies disrupted astrocyte–neuron adhesion in vitro but not astrocyte-astrocyte adhesion. Co-purification with AMOG suggested a neuronal receptor binding to AMOG is likely among these co-purified proteins. Subsequent studies demonstrated AMOG-containing liposomes could adhere to cultured granule cells, leading to the cloning of the AMOG gene and its identification as a β-subunit isoform of NKA, renamed AMOG/β_2_ [12,62].

The cloning of AMOG/β_2_ [11,78] allowed further investigation of its function in transfected cells. Müller-Husmann et al. [79] used AMOG-transfected L-cells as substrates for cerebellar neuron neurite outgrowth, observing increased neurite length—a response inhibited by antibodies against AMOG/β_2_. The soluble recombinant extracellular domain of AMOG/β_2_ partially blocked neurite outgrowth on AMOG/β_2_-expressing L-cells, while L-cells transfected with the mouse β_1_ subunit did not affect neurite extension. These findings indicate that AMOG/β_2_ interacts with an unidentified neuronal receptor to enhance neurite growth, likely through signal transduction pathways.

To elucidate AMOG’s physiological role, β_2_ knockout mice were generated. At postnatal day 15, these mice exhibited motor coordination abnormalities, tremors, and paralysis, leading to death by days 17–18 due to dysfunction in critical brain structures [80]. Morphological analyses revealed enlarged ventricles and swollen astrocytic end feet, yet several brain regions appeared normal. This raises questions about AMOG/β_2_’s potential role in brain development. Interestingly, AMOG/β_2_ antibodies interfere with granule cell migration in vitro [10], suggesting other molecules might compensate for the lack of AMOG/β_2_ in AMOG^(−/−)^ mutants.

To differentiate pump activity from adhesion molecule function, Weber et al. [81] generated β_2_/β_1_ knock-in mutant mice, where β_1_ expression replaced β_2_ expression. Unlike β_2_-deficient animals, these knock-in mutants had a normal lifespan and did not exhibit swollen end feet. Photoreceptor cell degeneration was reduced compared to β_2_ null mutants, suggesting the β_1_ subunit can partially substitute for β_2_ function. However, the role of AMOG/β_2_ as an adhesion molecule remains to be fully elucidated.

Exploring the long-term consequences of AMOG/β_2_ deficiency, Isenmann et al. [82] grafted parts of the embryonic telencephalic anlage from deficient mice into wild-type mice, analyzing up to 500 days post-transplantation. The grafts developed normally, forming solid neural tissue indistinguishable from controls. No signs of degeneration were observed.

By the late 1990s, the understanding of AMOG/β_2_ remained incomplete. It had been identified as a glial adhesion molecule, prominently expressed by Bergmann glial cells during cerebellar granule cell migration and in astrocytes of the mature cerebellum. AMOG/β_2_ was also detected in cerebellar granular cells and was recognized as a heterophilic adhesion molecule interacting with an as-yet-unidentified neuronal receptor that played a critical role in brain development. Studies involving β_2_ knockout mice showed that these animals died shortly after birth, exhibiting severe neurological deficits, while β_2_/β_1_ knock-in mice survived. This survival in knock-in mice highlighted that the observed deficits were linked to altered Na^+^ and K^+^ pumping, which directly affected neuronal activity. Despite these findings, the identity of the specific neuronal receptor for AMOG/β_2_ remains unknown, leaving a key aspect of its function unresolved.

## 3. Current Understanding and Knowledge Gaps

### 3.1. β_1_-Subunit as a Homophilic Adhesion Molecule in Epithelia

Numerous studies have shown that the β_1_ subunit functions as a homotypic cell adhesion molecule. Its interaction with neighboring cells is essential for the lateral polarization of the pump and the apico–basal polarization of epithelial cells (Figure 1). Epithelial cells deficient in the β_1_ subunit undergo epithelial–mesenchymal transition (EMT), but transfection with the β_1_ subunit restores their adhesive properties and enhances polarization [15].

At the start of the 21st century, research on epithelial cells highlighted the role of the β_1_ isoform of NKA, linking β_1_ subunit expression to cell–cell adhesion and cell polarity [18,49]. Further studies established that the β_1_ subunit acts as a homotypic cell adhesion molecule, with its interaction critical for both the lateral polarization of the pump [16] and cell–cell adhesion [17,19,21]. Molecular research identified a specific 10 amino-acid sequence (198–207) crucial for trans-dimerization of β_1_ subunits and cell–cell adhesion. This sequence explains the lack of stable interaction between β_1_ subunits from different species, such as rats and dogs [20]. Using in silico methods, residues at the interface of β_1_-subunits from dog epithelium were identified, confirming their role in protein–protein interactions through site mutagenesis [83].

### 3.2. Gaps and Unresolved Questions

Despite progress in understanding AMOG/β_2_, several gaps and questions remain about its roles and interactions in the nervous system. If AMOG/β_2_ is a heterophilic adhesion molecule, identifying its receptor molecule in neurons remains a significant gap. Early studies by Schachner’s group attempted to determine if known adhesion molecules like L1, NCAM, or NAG were the receptors, but none were identified [10,11]. Although many new CAMs have been identified in the CNS, none appear to interact with AMOG/β_2_.

Research suggests the interaction between AMOG/β_2_ and its elusive neuronal receptor may activate pathways promoting neurite outgrowth and cell migration. Some studies have linked specific signaling pathways to AMOG/β_2_ [84,85,86]. However, these generally assume activation through an unidentified receptor without identifying it. Litan and coworkers [86] demonstrated that cerebellar granule cells express α_1_, α_2_, β_1_, and β_2_ isoforms during postnatal differentiation. They proposed a model where AMOG/β_2_ activates the mTOR/p70S6 kinase pathway, which is associated with cell size regulation, though it remains unclear whether the neighboring cell is another granule cell or an astrocyte expressing AMOG/β_2_ [86]. Antonicek et al. [10] suggested that AMOG does not facilitate astrocyte adhesion, ruling out homophilic interactions. Yet U87-MG glioma cells transfected with AMOG/β_2_ form aggregates, indicating AMOG/β_2_ can engage in homophilic interactions [84]. This raises the following question: if AMOG can engage in homophilic interactions, why can they not occur in astrocytes?

**Figure 1 ijms-26-08744-f001:**
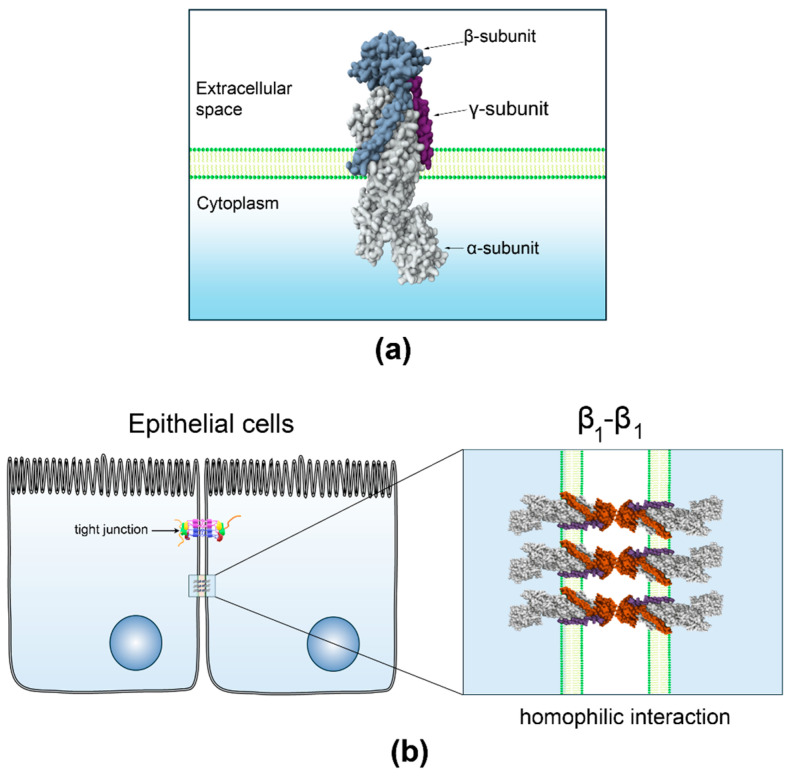
β_1_ subunit functions as a homophilic cell adhesion molecule in epithelia. (**a**) Three-dimensional structure of NKA, created from PB ID 7WYT [87] with Mol* Viewer [88]. (**b**) Homophilic and species-specific interaction of β_1_ subunits of NKA localized at the lateral membrane of neighboring epithelial cells.

Zlokovic et al. [89] identified β_1_ and β_2_ isoforms in rat cerebral microvessels. Boer et al. [90] described AMOG/β_2_ expression in human cerebral cortex development, showing high perivascular abundance and glial end feet staining, suggesting a role in vascular integrity. This prompts the question of whether AMOG/β_2_ facilitates adhesion at the blood–brain barrier between astrocytes and endothelial cells via homophilic interactions.

Observations from AMOG/β_2_ knockout mice reveal they do not survive, displaying motor incoordination by postnatal day 15 and dying by days 17–18. Interestingly, their cerebella appear unaffected, with cortical layer thickness like that of wild-type mice. This suggests AMOG/β_2_ may not be essential for early cell migration. Instead, its role may be in maintaining the osmotic balance for neuronal activity [80]. This raises a crucial question: how can we differentiate between these two functions of AMOG/β_2_?

## 4. Recent Findings

Although the neuronal receptor for AMOG/β_2_ remains unidentified, studies suggest that AMOG/β_2_ may also function as a homophilic cell adhesion molecule. Recent research hypothesizes that AMOG/β_2_, like the β_1_ subunit in epithelial cells, could mediate homophilic interactions. Given the distinct expression patterns of NKA subunits [91,92], both homophilic and heterophilic interactions at the CNS are plausible. Thus, the β_2_ subunit may engage in heterophilic interactions with neurons at tripartite synapses (Figure 2a), in homophilic interactions with cerebellar granule cells during migration (Figure 2b), and in homophilic interactions with endothelial cells at the blood–brain barrier (Figure 2c) [30]. Roldán et al. demonstrated that YFP-tagged AMOG/β_2_ in CHO and MDCK cells facilitates aggregation through β_2_-β_2_ interactions [93] (Figure 3). Dilution experiments confirmed that aggregation was proportional to β_2_ expression, though heterotypic interactions were also considered. Pull-down assays revealed no interaction in vitro; however, co-culture experiments detected β_2_-β_2_ interactions, emphasizing the importance of cellular context [93]. Cerebral organoids may be an effective model for studying such interactions in vivo. β_1_ subunit homophilic interactions are well-documented [21,50] with species-specific residues critical for adhesion [20,83]. Roldán et al. [93] and Ramírez-Salinas et al. [94] identified glycosylated extracellular domains in AMOG/β_2_ forming likely homodimers, though β_2_-β_2_ interfaces were smaller and weaker compared to β_1_-β_1_. Molecular modeling and molecular dynamics (MD) simulations revealed key residues in β_1_-β_1_ dimers (e.g., Gly225 and Leu266) and distributed hot spots in β_2_-β_2_ dimers. Binding free energies were calculated as −22,671.13 kcal/mol for β_1_-β_1_ and −19,707.5 kcal/mol for β_2_-β_2_ dimers. Glycosylation plays a vital role in stabilizing β_1_-β_1_ and β_2_-β_2_ interactions. While β_1_ has three conserved glycosylation sites, β_2_ contains four additional sites, totaling seven (Figure 4). These glycosylation sites are essential for adhesion, with distinct contributions in β_1_ and β_2_ dimers. For instance, Asn265 in β_1_-β_1_ mediates intramolecular interactions, while Asn153 in β_2_-β_2_ facilitates both intra- and intermolecular contact [93,94].

## 5. Looking for the Partner

The hypothesis that AMOG/β_2_ requires a specific neuronal partner or receptor is supported by evidence suggesting its adhesive function may depend on a heterophilic interaction with a neuronal surface protein. While some studies have indicated potential homophilic β_2_-β_2_ interactions, especially in heterologous systems like CHO and MDCK cells [93], such binding has not been observed between astrocytes in vivo [10], or in biochemical assays with purified proteins. This discrepancy suggests that AMOG/β_2_ may need a specific neuronal binding partner, which is absent in astrocytes but present at the neuron–astrocyte interface during CNS development or maintenance.

### 5.1. Experimental Strategies

To identify the elusive AMOG/β_2_ receptor involved in neuron–astrocyte adhesion, several complementary experimental strategies can be pursued. 

Co-Immunoprecipitation and Mass Spectrometry: This technique allows for the isolation of protein complexes associated with AMOG/β_2_ from astrocyte–neuron co-cultures or brain membrane fractions. By immunoprecipitating AMOG/β_2_ under native conditions and analyzing co-precipitated proteins, potential binding partners may be identified. However, this method can miss transient or low-affinity interactions, especially those involving membrane proteins. Careful optimization of detergents is required to preserve native protein structures [95].

Proximity Labeling Techniques (BioID or APEX): These methods involve fusing AMOG/β_2_ with an enzyme that biotinylates neighboring proteins in live cells. When expressed in astrocytes or neuron–astrocyte co-cultures, these tags label proteins near AMOG/β_2_, which can then be purified using streptavidin beads and identified by mass spectrometry. This approach is particularly useful for capturing transient interactions in their native membrane environment, providing insights into the proteins that interact with AMOG/β_2_ in vivo [96,97].

Functional Screening with Adhesion Assays: This strategy involves expressing AMOG/β_2_ in non-adhesive cells, such as CHO or HEK cells, and co-culturing them with a membrane protein cDNA library derived from neurons. Cells that adhere selectively can be isolated, and the corresponding cDNA identified, providing direct functional evidence of interaction. This method can reveal novel interaction partners that mediate AMOG/β_2_-dependent adhesion.

Genome-Wide CRISPR Knockout Screens: By employing CRISPR technology in neurons, researchers can identify genes required for adhesion to AMOG/β_2_-expressing astrocytes. Loss of adhesion in co-culture systems would suggest that the disrupted gene encodes a critical component of the receptor complex. This unbiased approach can uncover novel interactors that might not be predicted based on sequence homology or prior knowledge [98].

In Situ Chemical Crosslinking: Applying membrane-permeable crosslinkers to live neuron–astrocyte cultures can stabilize protein complexes in their native context. Following crosslinking, AMOG/β_2_ complexes can be immunoprecipitated and analyzed via proteomics. This method provides a snapshot of direct physical interactions that occur at the cell surface in vivo [99].

FRET or Bimolecular Fluorescence Complementation (BiFC): If specific candidate receptors are available, these techniques can be used to assess interactions in live cells. They rely on fluorescent signal restoration when two halves of a split fluorophore are brought together by the interaction of two fused proteins, enabling spatial and temporal visualization of the interaction [100].

To confirm the physiological relevance of any identified candidates, cerebral organoids can be employed. These three-dimensional structures, derived from human pluripotent stem cells, mimic aspects of the human brain’s architecture and function. They provide a valuable model for studying brain development, disease mechanisms, and drug responses. Cerebral organoids exhibit layered structures similar to those found in the developing brain, including regions resembling the cortex, hippocampus, and other areas [101,102]. They contain neurons, astrocytes, and other glial cells, allowing for the study of intercellular interactions. Knockouts of the putative receptor can be used to determine whether loss of the candidate protein disrupts astrocyte–neuron adhesion in vivo, mimicking the phenotype observed in AMOG/β_2_-deficient animals. This comprehensive approach, ranging from biochemical isolation and proximity labeling to functional genetic screens and in vivo validation, forms a robust framework for identifying the receptor or complex responsible for AMOG/β_2_-mediated adhesion in the central nervous system.

### 5.2. Candidates for AMOG/β_2_ Receptor

Two intriguing candidates for the AMOG/β_2_ receptor are Tetraspanin 31 (TSPAN31) and Reticulon 4 (RTN4/NogoA). Both have emerged from protein–protein interaction databases [103,104].

TSPAN31: As a member of the Tetraspanin family, TSPAN31 is notable for its role in organizing membrane microdomains and mediating lateral interactions between cell surface proteins. Tetraspanins act as molecular scaffolds, clustering adhesion molecules, integrins, and signaling receptors into functional complexes [105,106,107]. TSPAN31 has been implicated in cell adhesion, migration, and membrane signaling functions aligning closely with AMOG/β_2_ activities. TSPAN31 might associate in cis with a neuronal adhesion receptor, creating a complex that interacts in trans with AMOG/β_2_ on astrocytes. Alternatively, TSPAN31 might directly stabilize or present the neuronal partner required for AMOG/β_2_ recognition. Its potential involvement raises intriguing questions about how these microdomain organizations contribute to AMOG/β_2_ function.RTN4 (Nogo-A): Known for inhibiting neurite outgrowth, RTN4 has a complex topology and is present not only in the endoplasmic reticulum but also on the plasma membrane of axons and dendrites [108]. Its interactions with membrane proteins suggest it could serve as a scaffold or modulator for a receptor complex capable of interacting with AMOG/β_2_. If RTN4 is enriched in specific neuronal compartments, such as dendritic spines or axon terminals, its spatial distribution could explain the specificity and context-dependence of AMOG/β_2_-mediated adhesion during synaptogenesis or glial ensheathment. RTN4’s role in membrane dynamics and its interaction network make it a compelling candidate for further investigation.

Together, TSPAN31 and RTN4 represent distinct but not mutually exclusive modes of interaction with AMOG/β_2_: one through microdomain organization and lateral associations (TSPAN31), and the other via scaffolding or presenting transmembrane receptors (RTN4). Both could act directly or indirectly to facilitate the molecular handshake between astrocytes and neurons. Elucidating whether either—or both—of these proteins interact functionally with AMOG/β_2_ will require targeted biochemical, cellular, and in vivo studies. Nonetheless, their candidacy reinforces the emerging view that AMOG/β_2_ operates not as a lone adhesion molecule but as part of a larger multiprotein complex that coordinates glia–neuron communication at the cell surface.

## 6. Functional Implications in the CNS

The implications of AMOG/β_2_-mediated adhesion in the CNS extend beyond its role in NKA. Positioned in astrocytes, AMOG/β_2_ influences neuron–glia communication, contributing to processes such as synaptic organization, neurogenesis, and potentially pathological conditions like glioblastoma.

A key role of AMOG/β_2_ is facilitating communication at the tripartite synapse. Astrocytes are crucial for synapse formation, maintenance, and plasticity, interacting closely with neuronal components [70,71,72,73]. AMOG/β_2_ may enhance the alignment of astrocytic processes with synaptic sites, optimizing neurotransmitter uptake, ion homeostasis, and gliotransmitter release. Its involvement in metabolic coupling [109] suggests that it plays a role in supporting synaptic interfaces.

During neurogenesis, AMOG/β_2_-mediated adhesion could aid neuronal migration and synaptic connection formation, anchoring developing neurons to astrocytic scaffolds. Disruption in this mechanism might lead to aberrant synapse formation, linking AMOG/β_2_ to neurodevelopmental disorders. In glioblastoma, changes in adhesion molecules often affect tumor invasiveness and glial cell transformation. AMOG/β_2_ dysregulation could influence tumor progression, possibly facilitating cell detachment and invasion or altering tumor–microenvironment interactions [110,111,112,113]. Beyond glioblastoma, AMOG/β_2_ could impact other neuropathological conditions [90,111]. Impaired function may disrupt astrocytic support in neurodegenerative diseases, affecting synaptic stability and metabolic coupling. Altered expressions or interactions might also influence neuroinflammation, in which astrocytes play a key role.

AMOG/β_2_ acts as a regulator of cell–cell communication and structural organization in the CNS. Future research to identify its neuronal receptor(s) and signaling pathways is essential for understanding its influence on neuronal function and CNS health. These mechanisms could reveal therapeutic targets for neurological disorders and gliomas involving adhesion dynamics.

## 7. Future Directions

### 7.1. Research Avenues and Implications

Identifying the binding partner(s) of AMOG/β_2_ opens several promising research directions. In neurodevelopment, understanding how AMOG/β_2_ mediates neuron–glia interactions could illuminate mechanisms guiding neuronal migration, circuit formation, and synaptic stabilization. Insights into these processes may clarify how changes in adhesion dynamics contribute to developmental disorders such as autism spectrum disorders or cortical malformations [114]. Understanding these pathways could lead to interventions that stabilize synaptic connections and support proper neural circuit formation.

In neurodegeneration, where synaptic loss and glial reactivity are significant pathological features, altered AMOG/β_2_ expression or function might influence disease progression. Investigating its role in maintaining neuron–glia contact could shed light on early events in diseases like Alzheimer’s or ALS, where supportive glial functions become compromised [115]. Understanding these interactions may offer new strategies to preserve synaptic integrity and delay neurodegenerative processes.

In cancer, particularly glioblastoma, the progressive loss of AMOG/β_2_ is associated with increased tumor invasiveness [85,111]. Uncovering its adhesion partners may help define the molecular switch from organized glial tissue to a disaggregated, migratory tumor phenotype. This could lead to identifying biomarkers for tumor progression or new molecular checkpoints vulnerable to therapeutic intervention, potentially improving prognosis and treatment options.

### 7.2. Therapeutic Targeting Potential

Studying AMOG/β_2_’s adhesion mechanisms could inspire novel therapeutic strategies. For instance, developing mimetics or stabilizers of AMOG/β_2_-mediated adhesion might restore normal glia–neuron or glia–endothelial interactions in degenerative diseases. By enhancing these interactions, it may be possible to improve neuronal support and function. Conversely, in glioblastoma, artificially reactivating AMOG/β_2_ expression or mimicking its adhesive function might reduce invasiveness or sensitize tumor cells to treatment, potentially inhibiting tumor growth and spread. This approach could lead to therapies that specifically target the adhesive properties of tumor cells, reducing their ability to migrate and invade surrounding tissues.

Targeting the β_2_ subunit and its receptor pair could also represent a new class of adhesion-modulating therapies, which will be especially relevant to CNS conditions in which adhesion loss precedes or drives pathology. As AMOG/β_2_ bridges ion transport and adhesion signaling, it offers dual-entry points for therapeutic modulation—both electrochemical and structural. This dual functionality presents unique opportunities for designing interventions that can modulate both cellular adhesion and ion homeostasis, addressing multiple facets of CNS disorders (electrochemical and structural) simultaneously.

## 8. Summary

The β_2_ subunit of the Na^+^/K^+^-ATPase (NKA), also known as AMOG/β_2_, integrates dual roles in ion transport and cell adhesion. Initially characterized for its ability to promote neuron–glia adhesion during development, AMOG/β_2_ has since emerged as a key regulator of central nervous system (CNS) physiology and pathology. Despite its well-established adhesive function, the identity of its binding partner(s) in the CNS remains elusive, constituting the central question addressed in this review.

Elucidating the molecular partners of AMOG/β_2_ is crucial to understanding how adhesive interactions shape neurodevelopmental processes. Such insights may also clarify how disruptions in these interactions contribute to CNS disorders, including neurodegeneration and glioblastoma. Notably, AMOG/β_2_ expression is reduced in high-grade gliomas, supporting its role as a malignancy suppressor and a marker of glial differentiation.

Adhesion molecules like β_2_ are essential for organizing CNS architecture, orchestrating cell–cell interactions, modulating signaling pathways, and adapting to developmental and pathological cues. Investigating AMOG/β_2_ may therefore advance our understanding of neurobiology and reveal novel therapeutic targets.

Identifying AMOG/β_2_’s binding partners and elucidating its mechanisms of action may ultimately reshape our understanding of structural connectivity and signal integration in the CNS, with implications for brain development, plasticity, and disease.

## 9. Concluding Remarks

Understanding AMOG/β_2_’s role in the CNS opens up new research avenues in neurodevelopment, neurodegeneration, and glioma biology. Identifying its binding partners will clarify its function in neuron–glia interactions and its disruption in disease. As a dual-function target, AMOG/β_2_ offers potential therapeutic strategies to modulate adhesion in developmental disorders, neurodegenerative diseases, and brain tumors.

## Figures and Tables

**Figure 2 ijms-26-08744-f002:**
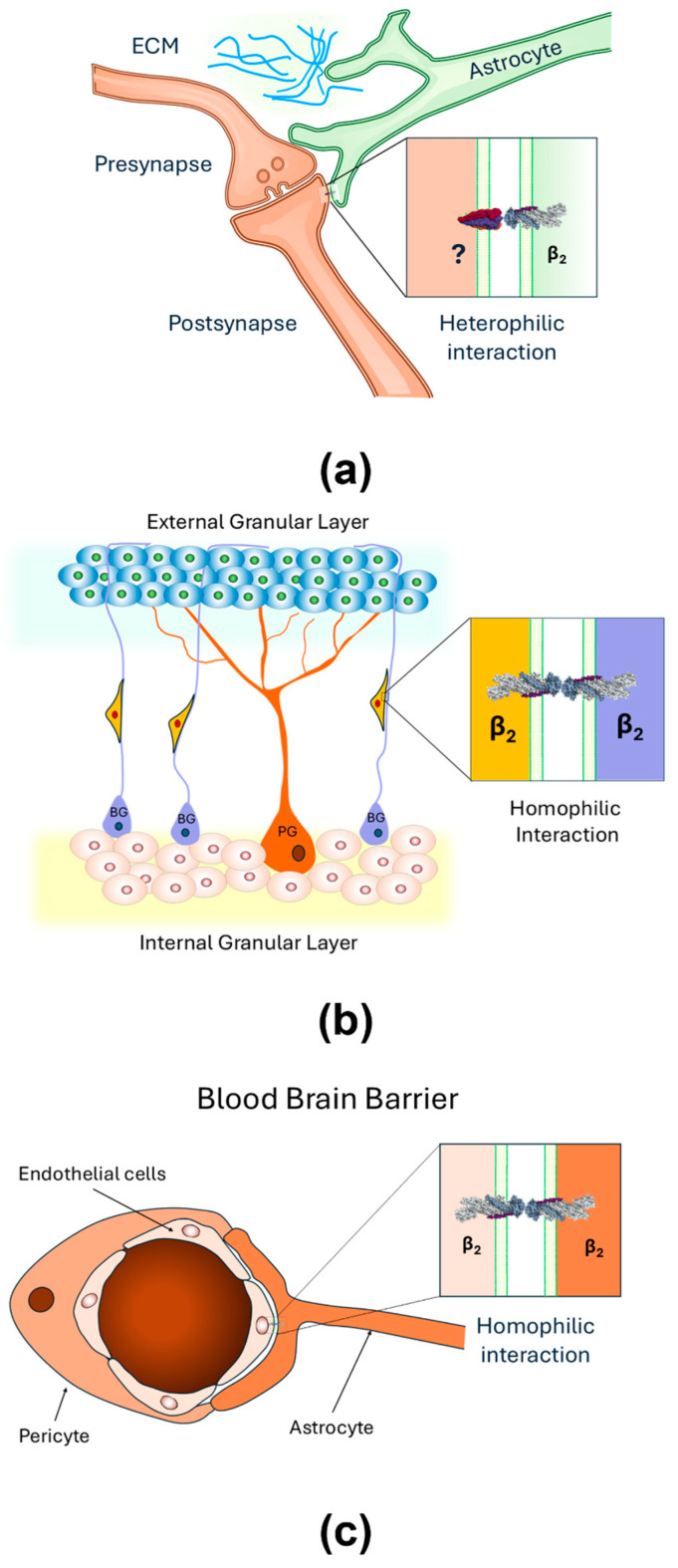
Homophilic and heterophilic interactions of AMOG/β_2_ are plausible at the CNS. (**a**) At tripartite synapses, astrocytic processes may contact neurons through heterophilic interactions between astrocytic AMOG/β2 and an unidentified neuronal partner (?); (**b**) AMOG/β_2_ may engage in homophilic interactions with cerebellar granule cells during migration; (**c**) AMOG/β_2_ may engage in homophilic interactions with endothelial cells at the blood–brain barrier.

**Figure 3 ijms-26-08744-f003:**
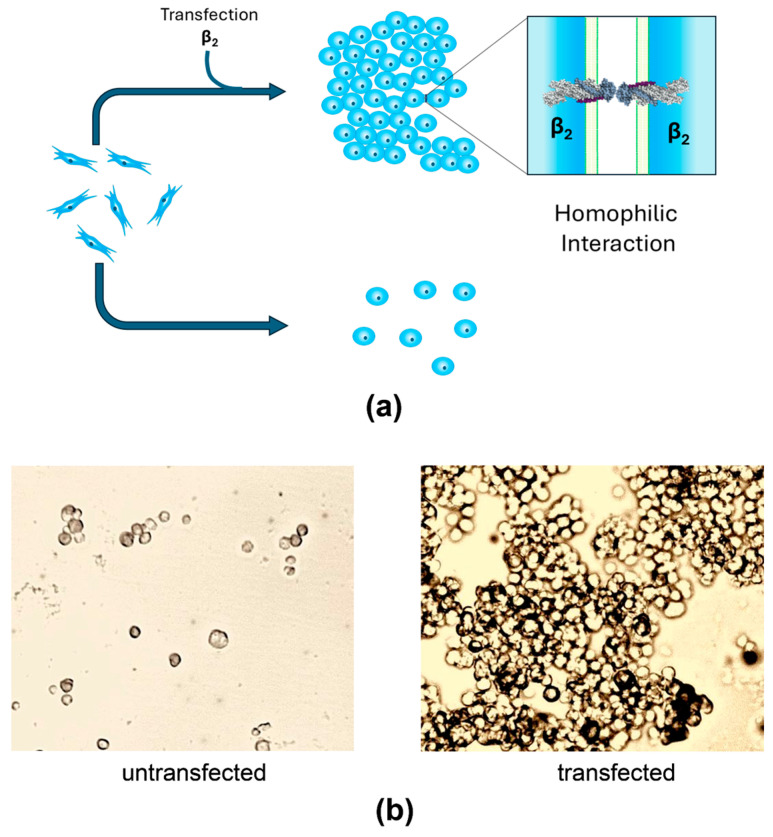
AMOG/β_2_ transfected in CHO fibroblasts facilitates aggregation through β_2_-β_2_ interactions. (**a**) A simple illustration of the aggregation assay of transfected or untransfected CHO cells. (**b**) Images taken by light microscopy of dispersed cells of untransfected CHO fibroblast (left) and big aggregates of AMOG/β_2_ transfected CHO cells (right). Adapted from Roldán et al. (2022) [93].

**Figure 4 ijms-26-08744-f004:**
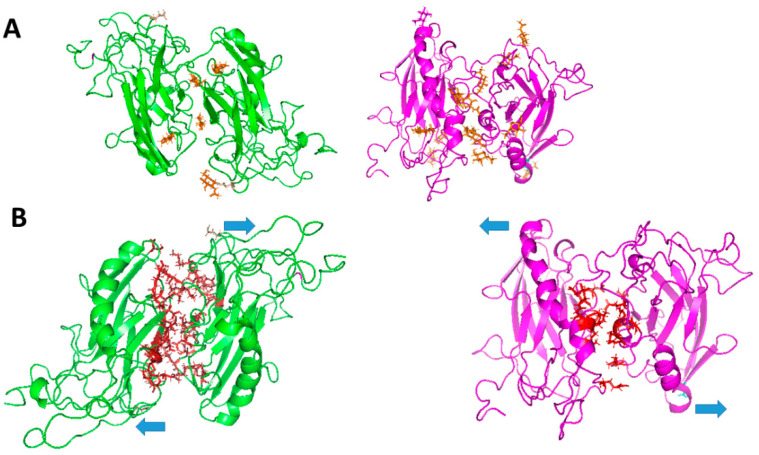
Three-dimensional models of trans-dimers of β_1_ and β_2_ subunits of human Na^+^/K^+^-ATPase. (**A**) N-Glycosylation sites of β_1_ and β_2_ dimers. N-Glycosylations are depicted in orange. (**B**) Protein–protein interfaces of β_1_ and β_2_ trans-dimers. The protein interface is highlighted in red in both models. The 3D models of β_1_ and β_2_ are shown in green and magenta, respectively. Blue arrows denote the orientations of the two β-subunits comprising the dimer, in trans. Modified from Ramírez-Salinas et al. [94].

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
