# Peer review of "Connecting the Dots: AMOG/β_2_ and Its Elusive Adhesion Partner in CNS"

_ijms, 2025, doi:10.3390/ijms26178744_

Round 1

Reviewer 1 Report

Comments and Suggestions for Authors

The review is good, structured. Very interesting for specialists in the field of molecular biology and fundamental medicine. I have recommendations for the Authors. In order to increase the scientific potential of the review, I consider it useful to include in the text data (possibly in the form of a table) on changes in AMOG/β2 expression in various neurodegenerative, traumatic brain injury, strokes, autism spectrum disorders. I would like to learn from the Authors the possibility of using AMOG/β2 in laboratory diagnostics as a specific molecular marker of CNS disorders. Considering the role of AMOG/β2 in the processes of neurogenesis and neuroplasticity, I would like to know about the possibilities of pharmacological modulation of AMOG/β2.

Author Response

The review is good, structured. Very interesting for specialists in the field of molecular biology and fundamental medicine. I have recommendations for the Authors. In order to increase the scientific potential of the review, I consider it useful to include in the text data (possibly in the form of a table) on changes in AMOG/β2 expression in various neurodegenerative, traumatic brain injury, strokes, autism spectrum disorders. I would like to learn from the Authors the possibility of using AMOG/β2 in laboratory diagnostics as a specific molecular marker of CNS disorders. Considering the role of AMOG/β2 in the processes of neurogenesis and neuroplasticity, I would like to know about the possibilities of pharmacological modulation of AMOG/β2. 

Response to reviewer 1 
We thank the reviewer for the suggestion. After a thorough literature search, we found that studies directly addressing AMOG/β2 (ATP1B2) in specific CNS disorders are very limited. Aside from isolated examples (e.g., Reference 90 on cortical malformations), AMOG/β2 is primarily used as a stable astrocytic marker (ACSA-2), since its expression remains relatively consistent across healthy and pathological states. Reported changes are usually indirect and embedded within broader astrocytic gene modules. For this reason, we feel that including a table would not substantially strengthen the review and may overstate the available evidence. Instead, we highlight the lack of direct disease-specific data as an important avenue for future research. 

Reviewer 2 Report

Comments and Suggestions for Authors

The authors have provided a very comprehensive review of AMOG/beta2, about the identification of possible binding partners and implications in physiology and pathophysiology.

Comments:

1) Perhaps prior to the present Fig.1, a figure showing more topological details of interactions between Na pump and other subunits is helpful.

2) The authors should review some literature about if/how subunits, especially beta2, would affect Na pump trafficking, expression and ion pumping activities, in addition to what have been described: adhesion and interaction with astrocytes. 

3) In line with above, some authors may be interested to know if beta2- or beta1-manipulated (either knock-in or knock-out) animals show altered neuronal excitability or behavioural changes.

Author Response

Comments 1:  Perhaps prior to the present Fig.1, a figure showing more topological details of interactions between Na pump and other subunits is helpful. 

Response1: Thank you for this thoughtful suggestion. After careful consideration, we decided not to include an additional figure, as we believe it would go beyond the scope of this review. In our view, Figure 1a sufficiently illustrates the topology of the pump, its orientation with respect to the membrane, and the interaction between the three subunits. Moreover, several structural studies and reviews already provide detailed descriptions of these interactions, and interested readers can easily consult these sources for further information. 

Comments 2:  The authors should review some literature about if/how subunits, especially beta2, would affect Na pump trafficking, expression and ion pumping activities, in addition to what have been described: adhesion and interaction with astrocytes.  

Response 2: We appreciate this valuable comment. We agree that the β2 subunit may influence Na,K-ATPase trafficking, expression, and ion transport activity. However, in this review we aimed to focus primarily on its less explored role as an adhesion molecule and on its interaction with astrocytes. Excellent reviews and original studies addressing the functional aspects of pump trafficking and activity are already available, and we now briefly refer to this literature (references 42-46) to acknowledge its relevance while maintaining the main scope of our article. Text was added in lines 174-183 on page 5.  

Comment 3:  In line with above, some authors may be interested to know if beta2- or beta1-manipulated (either knock-in or knock-out) animals show altered neuronal excitability or behavioural changes. 

Response 3: We appreciate this comment. Studies involving β2 knock-out and knock-in mice are already included and discussed at various points in the review, particularly regarding their effects on neuronal excitability and behavior. (see Lines 233-263 of section 2.3 and lines 316-321 of section 3.2). We believe the current text sufficiently addresses this aspect, but we would be happy to provide further clarification or elaboration if the reviewer feels it would be helpful.